# Development of a Multivariate Model Focused on the Analysis of Water Availability in Mexico

Hugo Romero-Montoya *, Diana Sánchez-Partida, José-Luis Martínez-Flores and Patricia Cano-Olivos

Graduate Department of Logistics and Supply Chain, Universidad Popular Autónoma del Estado de Puebla (UPAEP University), Puebla 7241, Mexico; diana.sanchez@upaep.mx (D.S.-P.); joseluis.martinez01@upaep.mx (J.-L.M.-F.); patricia.cano@upaep.mx (P.C.-O.)
* Correspondence: hugo.romero@upaep.edu.mx

**Abstract:** The present study proposes developing a multivariate model that predicts water availability in Mexico through 26 variables related to aquifers, renewable water, demographic characteristics, rivers and basins, dams, and irrigation factors. The information inherent to them was extracted from the platform of the national water system using records from the 13 administrative hydrological regions between 2010 and 2017. The model is based on the multiple linear regression model and the variable selection method. The results show different versions of the model contrasted concerning the statistical assumptions of the multiple regression. Although the findings presented have implications in the development of strategies focused on a better distribution of the vital liquid, in the face of various projected scenarios based on the variables analyzed, it should be noted that the progressive improvement of the model was carried out through the use of techniques such as the transformation of variables, detection, and elimination of outliers. The final result is water availability in the face of various drought conditions explained by a model with 16 relevant variables. Said prediction model is helpful for the generation of drought mitigation strategies.

**Keywords:** multivariate model; SINA (National Water Information System); water availability; Hydrological Administrative Regions (RHA)

## 1. Introduction

Over the past quarter-century, the world has become more prone to droughts, and droughts are expected to be more widespread, intense, and frequent due to climate change [1]. At present, the effects of these have become a severe problem for water use in various sectors, for example, agriculture, the supply of drinking water and the production of hydroelectric energy, and adverse repercussions on ecosystems [2–4]. This natural phenomenon, intrinsically associated with the hydrological cycle, has been little studied and not precisely because it is unimportant. It is not very easy to analyze due to the multiple factors that can simultaneously cause and affect [5]. Of all-natural hazards, drought affects the most significant number of people worldwide, causing devastating impacts ranging from socioeconomic effects that start with crop failure to unemployment, erosion of assets and decreased income, and poor nutrition, which entails a reduction in risk absorption capacity, increasing the vulnerability of the communities that suffer from them and in general around the world [6,7].

Not surprisingly, droughts significantly affect livelihoods, reciprocally exacerbating poverty levels and sustainability for years to come [8]. Thus, the study of drought is becoming increasingly important, largely thanks to the magnitude and complexity of the effects in different environments. For its part, the implications of drought are much less evident than other climatic effects and extend over more expansive geographical areas than most of the other known dangers [9].

Drought has been gaining notoriety due to the damage it causes, which often has tremendous implications [10]. It is defined as a prolonged water scarcity period well below

normal levels, which causes adverse effects on production systems and resources [9,11]. Its definition has been the subject of numerous scientific studies, but climatic typologies' diversity makes it almost impossible to use the same threshold of rainfall deficit in two different places [12–14]. It avoids reaching a generalized and standard agreement on it. It should be noted that the definition of drought varies remarkably according to the geographic space on which it should be applied. Thus, the definitions are usually minimal since they do not establish reference thresholds and do not consider the climatic reality of the area affected by the rainfall deficit [15], which is why generally only conceptual names are taken into account.

As already mentioned, drought is a complex phenomenon to which it is difficult to give a specific approach, which includes all aspects and satisfies all expectations; drought is instead a peculiarity of the climate and the environment [13,16], which causes a water deficit low enough to have social, environmental, and economic effects, with short- and long-term repercussions [6]. Therefore, drought cannot be treated exclusively as a phenomenon of a solely physical nature, which is why it can be divided into four categories, meteorological, agricultural, hydrological, or socioeconomic [17]; said categorization depends to a large extent on the type of affectations that it causes in the place where it occurs and the territorial extension that it covers.

A hydrometeorological condition such as drought is characterized by unpredictable onset, duration, intensity, or severity in the territorial extension over which it occurs [18,19], causing lower-than-average water availability [20]. That is why, if there is a poor distribution of water during it, coupled with conditions such as overexploitation and pollution, the amount of liquid available for consumption is significantly reduced, that is, water is increasingly scarce, a situation that is alarmingly worse when there is a drought episode [21].

Different research studies have shown that the understanding of the availability of water can be improved as long as there is an appropriate perception of the phenomenon of drought, which is perhaps the first step to achieve adequate water management, mainly in times of scarcity, allowing to face the complications with much more success, and based on an appropriate allocation-balance of the water and the existing deficit [16]. However, a problem of this type requires considering that said natural hazard needs a specific policy that includes standardized precautionary and control measures [22] that avoid fostering the erroneous belief that after a drought, another will not occur the same or of more immense proportions [14].

To date, droughts are a prominent concern in the mapping and control of water resources [23]; their impacts are mainly the result of the interaction between natural events and the water demands of the regional socio-economy. Human activities also exacerbate these. Moreover, due to relationships, constant emphasis is placed on the fact that drought cannot be treated exclusively as a physical phenomenon [17]. However, it should instead be seen as a multiscale phenomenon or event in which the effects of deficit precipitation (meteorological drought) are evident in various systems (e.g., surface and groundwater hydrology, vegetation activity, and crop production) at various times scales over time [24].

As a consequence of these conditions, which directly influence competition for water resources, water is increasingly considered a scarce and valuable resource that demands rigor in its management and extreme care to avoid its misuse and waste [25]. There is currently growing concern about current practices' ineffectiveness in dealing with drought, which generally focuses on crisis management. Such practices are reactive and therefore only address the symptoms (impacts) of drought rather than the underlying causes of the vulnerabilities associated with those impacts [26].

It should be noted that drought is a unique natural hazard that causes damage to the ecology and the economy due to its multifaceted characteristics. Mitigating it requires a specific policy that takes into account precautionary and control measures [22].

That is why the formulation of policies and strategies on drought and water availability also represents a challenge, in which case it is much greater in vast countries such as India, China, Brazil, and the United States of America since drought rarely affects the entire

country at the same time, and mainly because the characteristics of drought vary in different parts of the country where diverse physiographic and climatic conditions are present [22].

On the other hand, in Mexico, a country with semi-arid conditions, the northern states cover 50% of the surface and present only 25%. In the narrow part of the country, representing 27.5% of the national territory, most rains are concentrated (49.6%); this region contains the south-southeast states [27].

It is worth mentioning that drought is a natural event that occurs recurrently in Mexico since most of its territory has characteristics of aridity and urban growth with inadequate consumption patterns [28,29]. Although part of the national territory has abundant rains, some regions do not have the vital liquid in the required quantities [5]. In Mexico, it is customary to define meteorological drought by its magnitude and duration based on monthly or annual rainfall data [12], so it might be thought that this type of forecasting method leaves out factors of another nature that could provide more information about the phenomenon.

According to some projections related to the Mexican XXI century, they emphasize the intensification of the pressures on natural resources essential for developing the economy, particularly on water. Particular attention should be paid to water stress and the effects that primarily focus on groundwater and surface water sources: That is why one in six aquifers located in the north and center of the country are currently compromised, overexploited, salinized, or contaminated [30]. In recent decades, droughts in Mexico are managed reactively, being treated the same as other "natural disasters" [4]. Shortly, Mexico could be exposed to very high-water stress in the basins of the Colorado, Bravo, and Lerma-Balsas rivers and the Baja California peninsula, the Valley of Mexico, and in the lagoon region, where the prognosis is alarming [16].

Given these conditions, it is essential to emphasize that not everything is negative; Mexico's experiences and studies on this subject show technical capacity to mitigate groundwater deterioration [16]. Although there are efforts in the matter of drought since 2010–2012, these change paradigms dealing with this phenomenon move from the reaction towards the preventive model oriented to risk management, called the National Program against Drought (PRONACOSE) [29]. This continued operating in the years 2014–2018. During this period, issues related to water security were addressed in the face of drought and floods. Since Mexico was considered highly vulnerable, the implementation of PRONACOSE became a valuable strategic resource to be prepared for any contingency [18].

For its part, the National Water Commission (CNA) has also been carrying out an unprecedented task for some time, to create a participatory water management system in the area of basins that currently covers the entire national territory [31].

Nevertheless, the lack of a national and regional drought policy framework, the limited coordination institutions, and the poor set of social impact indicators that make up the global early warning system remain ineffective in containing the effects of drought and the drought phenomenon in the country, since they only seem to focus on the political response [26].

It is vital to generate plans and strategies to overcome and mitigate its impacts, and adaptation and prevention are the best strategies. Without these principles, it is difficult to get out of a situation like drought [14].

However, the demand for water constitutes a fundamental piece of information that should not be forgotten when designing policies that impact water consumption. It also serves as a reference in the design, modernization, and operation of distribution systems [25].

Another important aspect to remember is that natural hydro-climatological processes and human activities are not a simple sum of both elements but complex feedback that gives rise to a hydrological system response, which must be analyzed under different perspectives [2]. Consequently, to solve this problem of multiple nature, modeling techniques that adequately represent the system and its change over time under different scenarios of

natural variation and intervention strategies that facilitate exploring various solutions and intervention strategies are required [32].

That is why, regarding aspects related to the integral management of water, all those factors that affect the process must be considered, such as environmental, social, legal, economic, and political [16], without leaving aside attempts to estimate predicted changes in meteorological parameters that directly or indirectly affect the frequency of drought episodes [23].

While the analysis and diagnosis have been approached by numerous investigations that focus on developing drought indicators adapted to different applications, the same does not happen concerning predicting the future. This aspect is not widely referenced [33].

The use of modeling with dome methods represents a resource used to determine the beginning and end of a drought, as well as in modeling with multiple variables and indices to combine information on drought from precipitation and soil moisture using the joint distribution function of the two variables [34]. Finally, it represents the explanation of the probabilities of drought and the return periods for the occurrence of events of this type in the next 5, 10, 20, 50, and 100 years in some regions [23].

Even today, there are still crucial obstacles in characterizing drought and interpreting the variables [23]. There are currently advances in identifying and using increasingly precise indicators of change, which will significantly boost progress in hydrologic drought assessment and prediction [35].

In general, in the analysis of drought, the priority task focuses on identifying an aspect or indicator that reflects the state of drought that exists at all times for a particular region. However, concerning the management of water resources, in addition to the said diagnosis of the evaluated conditions, a prediction of the state that the drought will take in the immediate future is required since this situation will essentially establish the type of measures to be adopted [33].

This study focuses on developing a multivariate model for explanatory and predictive purposes to analyze Mexico's water availability. The model was developed through progressive interactions in order to optimize its qualities. The variables used to cover most of the traits are primarily linked to the different types of droughts and Mexico's water availability. It should be noted that these variables have been grouped by their intention and their nature in the following categories: Meteorological, hydrological, geographical, development of water uses, as well as economic activities related to agriculture, industry, and the supply of water for domestic consumption or a significant deficit of water (critical value) both in time and space [36], without leaving behind the weights adapted concerning the drought index in the country and its socioeconomic properties [17]. All of this is in order to better understand the fundamental peculiarities concerning the availability of water. For this, it was necessary to use data to obtain the referents of each variable's behavior, extracted from the reports and records that the National Water System (SINA) facilitates through its online portal considering a period of 2010–2017.

In particular, the model tries to contribute to the knowledge base on the mitigation, evaluation, and multivariate attribution of droughts; the results obtained would suppose a contribution to the integrating global vision of a resource as vital as water and its adequate organization at the scale of hydrographic basins and aquifers in Mexico [14]. In general, critical key elements were analyzed and obtained regarding the variables correlated with the availability of water at the national level, which reveal significant elements to carry out a projection effective for the impact, including proactive risk management measures, plans of prevention to increase adaptive capacity, and efficient and effective emergency response programs in order to reduce the effects of droughts [1].

### 1.1. Literature Review

#### 1.1.1. Analysis of Drought and Water Availability

The causes of drought are unknown, but it is generally accepted to be due to alterations in atmospheric circulation patterns [16]. Therefore, it is necessary to underline

that man, in particular, can do little to prevent droughts; however, generally, there is the ability to minimize the effects through analysis and a better understanding of the phenomenon [36]. Because of this, the characterization of drought requires research focused on exhaustively analyzing and delineating trends and Spatio-temporal patterns more flexibly and comprehensively [23], and the proper management of water is also noticeable. Instead, it should include an understanding of the factors that regulate climate at the regional level, particularly regarding the hydrological cycle [37]. Given the exposed conditions, it makes sense to emphasize that, as more precise and multidimensional regional models are obtained, a much more exact starting point will be available to quantify water availability.

In other words, it is necessary to understand the hydrological cycle processes, and even more relevant to carry out the monitoring of the variables that make it up as well as their analysis, if accurate results are to be obtained about drought [38]. The complexity of drought and its broad impact components are also needed to adequately characterize the conditions that constitute the phenomenon's reality, for example, by integrating different variables or indices related to itself [13].

In short, once the relevant factors and triggers of the drought have been identified, the precision of the diagnosis will be substantially improved, which is essential for regional mitigation and the management of water resources. A good understanding of drought's persistent characteristics in a specific region is practical to understand its mechanisms [39] better.

Nevertheless, to carry out an adequate analysis of drought conditions generally requires integrating different variables or indices related to it [13]. Take as an example the specific case of a regional model in Oaxaca [38], which was based on 20 climatic and physiographic variables potentially helpful in predicting water consumption. Multivariate statistical techniques allowed them to obtain a regional model to predict annual average expenses to analyze surface water availability in the Mixteca region of Oaxaca and surrounding areas.

1.1.2. Prediction Models for Drought and Water Availability

Drought is one of the catastrophes capable of transforming the environment of a region on a large scale. The lack of water significantly deteriorates the inhabitants' quality and living conditions, causing environmental damage (flora, fauna, and landscape) [16]. It translates into a recurrent, atypical, and complex behavior, which no matter how particular it is, it is always possible to identify effects of variable intensity that in turn generate damage and impact on ecosystems, as well as in the availability of water resources in surface sources and underground [40].

In economic and social terms, the effects of drought affect the various sectors' effects. It is known more for its implications than for itself since the impacts caused by the lack of rain and water deficit are similar in any part of the world; of course, with their particularities [41].

Most of the research carried out in general focuses on atmospheric modeling [12]. For the most part, drought-related models almost always focus on a single variable (or indicator), which may not be sufficient because the phenomenon of drought is linked to multiple factors (for example, precipitation, runoff, and soil moisture) [34], but at the same time, most meteorological phenomena are multipliers. The advantages of using a multivariate analysis and evaluation strategy are evident [23] if we consider linear conjunction as a realistic option for solving the problem of combinatorial analysis of the different variables or drought indices, in order to integrate the information on it through various sources and characterize its conditions [13] through an optimal equation that assumes independent variables with which, in turn, an answer can be estimated [36].

Recent studies have shown that crucial obstacles still exist in characterizing drought and interpreting drought variables within the context of relevant information for regional monitoring, early warning, and planning of water resources. [23]. Undoubtedly, improvements have been made in the modeling of droughts in the last three decades. However,

more research is also needed to understand droughts' spatial and temporal complexity and climate change influences [35]. It would significantly help to develop support systems for decision-making to issue warnings promptly, assess risk, take precautionary measures, and find effective ways to develop and improve the flow of information from those responsible for these decision-making users of water [35].

On the other hand, research based on multivariate models for drought has focused primarily on developing multivariate indices. These constitute latent variables that belong to a particular physical meaning based on mathematical transformation (such as the difference or the relationship) of several variables [13]. It has resulted in different configurations that address the problem through distributions, forecasts, indices, water balances, and estimates [7,13,42,43]. So far, there are few references regarding developing a model focused on determining the relationship between a response variable, such as water availability, and the elements that make up the drought construct, being used as predictor variables for water availability.

The characterization of the demand for irrigation water from the original data recorded by the integrated water management system is a representative case of the use of error levels far removed from the statistical ranges of acceptance for multiple regression models, which presented problems of robustness and reliability of the system. These were identified in the data, which presented inaccuracies, errors, and noise [25]. On the other hand, in areas where water is too scarce to meet human activities' demand, evaluating this resource's availability is crucial in creating efficient optimal use strategies [38].

### 1.1.3. The Multivariate Model

More than one explanatory variable is used and this generates advantages when using more information in the model's construction and, consequently, making more precise estimates. Having more than one variable means that the coefficients are selected so that the sum of squares between the observed and predicted values is the minimum, that is, to minimize the residual variance [44].

The multivariate model represents a tool for developing research; its approach starts by examining and analyzing datasets. Linear regression is a statistical analysis that depends on modeling a relationship between two types of variables, dependent (response) and independent (predictor). The regression's primary purpose is to examine whether the independent variables successfully predict the outcome variable, and which independent variables are significant predictors of outcome [45].

A slope hypothesis test is performed to determine whether a predictor variable is related to the response variable. The null hypothesis states that the slope is zero, and the alternative hypothesis specifies that the slope is not equal to zero. If the test's $p$-value is less than $\alpha$, the null hypothesis is rejected, and it is concluded that the predictor variable is significantly related to the response variable [46].

### 1.1.4. The Assumptions and the Fit of the Model

The model's goodness is analyzed by studying the residuals' behavior, looking for unique or atypical observations with high "leverage" and points of influence. Observations that require particular attention are outliers: Those with huge residuals, unusual patterns, or individuals with significant influence on estimating the standard error of one or more βs [47]. These points require special attention because they could modify the regressor's center and make it biased, causing problems in the model's assumptions [45].

The multiple regression model's fundamental premises are summarized from three aspects primarily evaluated in the regression residuals: Linearity, which establishes that the relationship between the variables is linear; the independence, that is, the errors in the measurement of the explanatory variables are independent of each other; and finally, the non-existence of heteroscedasticity, which refers to the fact that the variance is not constant in the model [44]. In summary, in order to be able to draw highly valid conclusions concerning the proposed model or the phenomenon analyzed [48], any model seeks ho-

moscedasticity, that the errors have constant variance; normality, that the variables follow the normal law; and non-collinearity, that the independent variables are not correlated with each other [48].

### 1.1.5. Strategies to Correct for Adverse Effects on the Assumptions of the Model

Some transformations can be used to correct deviations from the model assumptions. For the non-homogeneity of the residuals, certain transformations can be tested: As a natural logarithm; square root if the distribution of the response variable is similar to that of Poisson; 1/the response variable if it is proportional to the predicted response variable. If linearity and homogeneity are maintained, non-normality does not matter if the sample size is large enough (n $\geq$ 50–100). Finally, if linearity is present, but not homogeneity, then the estimates of β are correct, but the standard errors can be corrected by calculating the "robust" ones [47]. The ways and methods to correct the deviations in the models' assumptions can be carried out through a discriminate observations' analysis so that only those that do not interfere or cause variations in the assumptions are considered.

### 1.1.6. Sequential Search Methods for Variables for the Model

Several selection methods for linear regression modeling specify how independent variables are entered into the analysis. Different regression models can be built from the same set of variables [47]. There are two types of sequential search approximations, the estimation by stages and the progressive and backward elimination; for each approximation, the variables are analyzed individually according to their contribution to predicting the response, adding or discarding according to their marginal contribution [49].

The estimate by stages (step by step): This method selects variables with the strongest correlation with the response variable to build a model later and test its significance according to what is stipulated by the researcher. It generally includes each variable before developing the equation so that each independent variable is selected based on its incremental contribution [49,50].

Progressive addition and elimination (forward and backward): This method is similar to that of stages; its main distinction lies in its progressive ability to add or eliminate variables at each stage. Generally, the method starts from predicting the predictor variables to a model with the response variable or from a model with all the response variables included, as the case may be. Once variables have been added or removed in the progressive addition and backward elimination schemes, there is no possibility of reversing this action [49,50].

The document has been organized into sections. The introduction presents the significant contextual aspects that shape the problem and current state of the prevailing conditions for drought and water availability in Mexico. Regarding the materials and methods, a description of the resources used to propose applying the applied methodology for developing the multivariate model is listed, which focuses on water availability in Mexico. Subsequently, the results obtained from the interaction of the methodological steps carried out are detailed, leaving room for the discussion section where the implications and findings that have been discovered from the results established in the previous sections are analyzed in detail. Finally, the conclusions reached from the critical factors revealed through joint development and research around the model's conception and conceptualization are presented.

## 2. Materials and Methods

This study is based on the characteristics identified by the geographical, social, and natural conditions of the Administrative Hydrological Regions (RHAs) that make up Mexico to establish causality between these factors and water availability. The factors considered are widely linked to the country's fundamental water issues and the hydrological cycle.

*2.1. The Geographical, Social, and Natural Features Characterize the Place of Study*

The Mexican Republic is located on the American continent within the northern hemisphere. Mexico is ranked 5th in territorial extension and 3rd in the number of its inhabitants. It is almost five times smaller than Canada, four times smaller than Brazil, and four and a half times smaller than the United States [51]. There is tremendous climatic and biological diversity in the subsoil's natural resources at the national level, with a marked differentiation in its territorial scope and a unique and essential political relationship with the United States of North America [52]. The great physiographic and climatic variety and its geographical location result in a large part of the territory's surface presenting arid and semi-arid characteristics due to its location in the desert strip of the northern hemisphere [14]. In Mexico, the usual succession of atmosphere states is the "dry" and rainy seasons, although their duration is relative for each region's climatic characteristics [53]. In the world ranking, Mexico is considered a country with low water availability. The country's most prosperous rivals in water availability are Canada and Brazil [27], which is why when compared at a global level, the national renewable supply places Mexico among the countries with a medium-high supply by its absolute values (25th), consequently with its underground resources, but medium-low in per capita terms (94th) (Food and Agriculture Organization, 2016). As with most natural resources, however, availability is not distributed equitably in a region, and in Mexico, it happens that renewable water is concentrated in the southern region, while in the center and north, it is relatively scarce [30].

*2.2. The SINA and the Data were Available to Deepen the Formation of the Variables Analyzed*

Considering the above, the registry of hydrological capacities in Mexico and its elements are primary tasks of the National and Regional Information Systems on quantity, quality, uses, and Water Conservation (SINA/SIRA's). The first is in charge of the National Water Commission, and the latter is in charge of the Basin Organizations. In turn, these aim to understand, in general terms, the information needs of federal, state, and regional institutions involved in water management as well as others linked to the water sector at the federal, state, municipal level, such as companies, teaching and research centers, NGOs, and other actors related to water; for example, water users, organized civil society, citizens in general, and international organizations [54].

For the most part, the data used to configure the different variables of the model were collected from the sections focused on the management of hydrological resources arranged in the records that the SINA maintains active in its open data platform, through which it was possible to form a total of 27 variables, including water availability. Each variable was grouped using a categorization established from 6 items: Aquifers, renewable water, demographic characteristics, irrigation factors, dams and rivers, and basins. This classification was devised to organize the variables extracted for analysis to associate the conditions of the different types of droughts meteorological, agricultural, hydrological and socioeconomic [17] with the variables and the items, as well as the ability to gain a better general understanding of the results obtained. It should be noted that these variables contain observations that were extracted from the records corresponding to a period between 2010 and 2017.

*2.3. The Proposed Methodological Design and Its Explanatory Scope*

The study design is based on a quantitative methodology with exploratory features applied mainly to analyze the variables. However, its orientation is primarily aimed at developing an explanatory approach by using multivariate statistical methods that indicate the development of a correlational investigation, which makes more sense about the causality to be demonstrated between the independent variables used and water availability. It should be noted that the proposed design is of the non-experimental transactional type since it is based on the collection of data during a specific period, so it was not necessary to use an analytical instrument to obtain the necessary observations and data. This consolidates the ex post facto orientation of the study since no variable was experimental

at any level or sense, but rather the conditions present in the variables during a specific period were taken into account, attributing to the proposed model an orientation based on the causality evidenced between the response variable and the independent variables, which points to a prospective type model. In particular, the proposed methodology seeks to facilitate the explanation of the causes that are related to a phenomenon as multifaceted as drought, since the interest of the model is focused on explaining why the phenomenon occurs and under what conditions it manifests itself or why it occurs in relation to two or more variables [55].

It should be noted that the proposed design is of the non-experimental transactional type since it is based on the collection of data during a specific period, so it was not necessary to use an analytical instrument to obtain the necessary observations and data. It consolidates the ex post facto orientation of the study since at no time were the variables experimented on under any level of treatment. Therefore, only those conditions that occurred in the variables during the period were taken into account, which gives the proposed model an orientation based on causality, which occurs between the response variable (availability of water) and the independent variables. This effect is translated into a prospective-type model, which generally projects a methodology that explains the implications of the multifaceted phenomenon of drought through the relationships between the variables and the conditions inherent to it [55].

For its part, the type of design proposed for this research is limited due to the impossibility of manipulating the independent variables, which requires an initial analysis of the dependent variable to later test the different independent variables under a retrospective approach [56]. The retrospective quality of the proposed approach indicates the use of historical information that supports the projection of a model based on those events that have taken place previously, and this dramatically facilitates obtaining the data for its manipulation [57]. To a large extent, the proposed design focuses on examining whether the changes in the response variable (water availability) are linked to the changes in the proposed predictor variables and how these occur naturally [58].

### 2.4. The Structure of the Proposed Methodology

The methodological structure focuses on developing a sequence of phases guided through different stages and steps to configure a multivariate model, in order to carry out the development of a series of activities such as the information gathering and integration of a database for the model. Figure 1 summarizes, in detail, the steps specified for the methodological sequence proposed in this study.

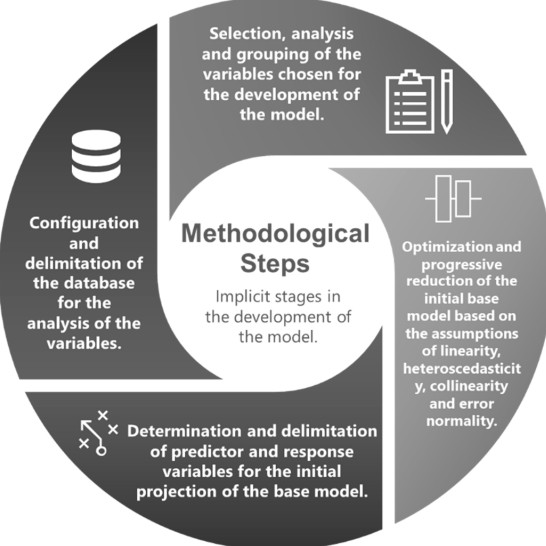

**Figure 1.** Proposed methodological model presented schematically, own elaboration.

*2.5. Selection, Analysis, and Grouping of the Variables Chosen for the Development of the Model*

Any data analysis aims to extract the raw data, the information, and the exact estimate [47]. The variables used and selected for the development of the model were determined from the availability of information, consistency of the data presented, and the relationship with the projected model's objectives. In general, SINA has different types of records related to various conditions directly linked to water resources management in the country and, more specifically, with the RHAs. Due to the number of records, the categorization of the variables mentioned above was very useful to the group and were differentiated based on time. This result of this process was 26 variables categorized according to 6 items with 104 observations collected in the years 2010–2017. Table 1 summarizes the variables obtained, the items they were categorized into, the description, and the code assigned to each of them.

**Table 1.** Categorization, classification, and definition of the 26 variables extracted from the SINA platform.

| Category | Variable | Code | Description |
|---|---|---|---|
| Aquifers | Availability (hm$^3$) | ADisp | The total volume of surface and underground renewable water that occurs naturally in a region. |
| | Weighted drought index | AIp | Average calculation stipulated from the conditions referred to by the records of the national drought monitor. |
| | Precipitation per year (sum mm) | APre | Precipitation is any product of the condensation of atmospheric water vapor-deposited on the Earth's surface. |
| | Deficit (hm$^3$) | ADef | The water deficit is, in short, a scarcity of water, so the concept is closely related to drought or scarcity. |
| | Average Recharge, Total Aquifers (hm$^3$) | ARmAquifer | Average Recharge is the average annual volume of water that enters an aquifer. |
| | Total of aquifers | ATa | Total number of underground structures that house water |
| | Total renewable (hm$^3$/year) | AaRt | Renewable water is called the maximum amount of water that is feasible to exploit annually in a country without altering the ecosystem, and that is renewed by the middle of the rain. |
| | Renewable water per capita (m$^3$/inhab/year) | AaRpc | The renewable water per capita of a country results from dividing its renewable resources by the number of inhabitants. |
| | Upper Average Natural Runoff Total (hm$^3$/year) | AEmst | It is the average annual volume of surface water that is captured by the hydrological basin itself |
| Renewable | Water Average renewable water (hm$^3$/year) | ARMed | Renewable water is called the maximum amount of water that is feasible to exploit annually in a country without altering the ecosystem, and that is renewed through the rain. |
| | Renewable water per capita 2030 (m$^3$/inhab/year) | ARP30 | Renewable water at the maximum amount of water feasible to exploit per inhabitant in the country projected to 2030. |

**Table 1.** *Cont.*

| Category | Variable | Code | Description |
|---|---|---|---|
| Demographic Characteristics | Population (Habs.) | CDPob | Set of inhabitants of the country. |
| | Continental Surface (km$^2$) | CDSc | The continental surface refers to the national territory's part articulated with the American Continent and the insular one to the country's islands' surface. |
| | Population Density (inhab/km$^2$) | CDdPob | Population density is equivalent to the number of inhabitants divided by the area where they live. |
| Irrigation Factors | Harvested area (ha) | FRSc | It is the area from which agricultural production was obtained. |
| | area (ha) | IrrigatedFRSr | The area of all the parcels that during the census year has been effectively irrigated at least once |
| | Number of users | FRUn | Several agricultural users per region. |
| | Production (thousands of tons) | FRProd | It is the result of the practice of agriculture. |
| | Yield (ton/ha) | FRRend | It is the ratio of the total production of a particular crop harvested per hectare of land used. |
| Dams | Capacity in Ordinary Maximum Water Level (NAMO) (hm$^3$) | PCna | Maximum Ordinary Water Level, the maximum level at which the dam can be operated to meet the demands, it can be drinking water, power generation, and/or irrigation |
| | Volume stored (hm$^3$) | PVa | It is the amount of water stored by the dams |
| | Number of dams | PNp | Several dams are currently existing. |
| Rivers and Basins | Natural runoff volume (hm3) | RCVen | It is the average annual volume of surface water captured by the hydrological basin's natural drainage network. |
| | Extraction volume (hm$^3$) | RCVext | Amount of water extracted. |
| | The average annual availability of rivers and basins (hm$^3$) | RCDma | It is the average annual volume of water s that, when positive, can be extracted from an aquifer for various uses, in addition to the extraction already granted and the natural discharge compromised, without putting in danger the balance of ecosystems. |
| | Area (km$^2$) | RCA | Territorial delimitation of the regions expressed in Km$^2$ |

*2.6. Configuration and Delimitation of the Data Structure for the Analysis of the Variables*

The analysis variables were structured in such a way to be able to form a database that was functionally useful for the development of the model. The observations of the 26 previously selected variables were ordered according to the year and the RHA of their origin. These 13 RHAs that make up the national territory are currently the water planning and management unit in the entire country [59]. In general, most of the records that the SINA provides for consultation are generalized under this classification. For their part, the RHAs are made up of basins groups, considered basic units for water resources management [60,61]. Therefore, the database structure is based on the 26 chosen variables, the projected analysis horizon, and the 13 RHAs that divide the national territory for its management, as shown in the map in Figure 2.

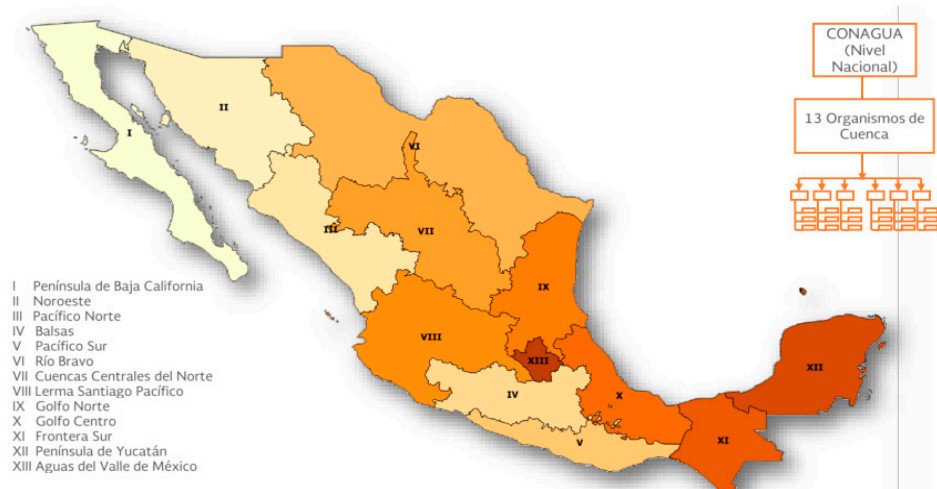

**Figure 2.** The geographical configuration of Mexico from the 13 RHAs for the management of water resources [62].

*2.7. Determination and Delimitation of the Predictor and Response Variables for the Initial Projection of the Base Model*

On the other hand, one of the most essential and common questions in the development of correlational research is whether there is a statistical relationship between a response variable ($Y$) and the explanatory variables ($Xi$) [46]. The multiple regression model is the extension of the simple regression model to $k$ explanatory variables. The multiple regression model structure is based on predicting and describing how $Y$ depends on $X$, which is expressed structurally in Equation (1) [46,47].

$$X_{1i},\ X_{2i},\ \ldots,\ X_{Ki}\ \rightarrow\ Y \tag{1}$$

The multiple regression examines how the independent variables successfully predict the response variable and which independent variables are significant predictors of the result from a linear combination of the values of one or more explanatory variables and an error term [45]. As shown in Equation (2), the linear regression model is established to evaluate said condition with more than one predictor variable.

$$\left\{ \begin{array}{c} Y = \beta_0 + \beta_1 X_{1i} + \beta_2 X_{2i} + \cdots + \beta_k X_{ki} + \varepsilon_i \\ E(\varepsilon) = 0,\ Var\ (\varepsilon) = \sigma^2 < +\infty \end{array} \right\} \tag{2}$$

The proposed multivariate model's initial projection was necessary to specify a response variable for its development. This specific case was to carry out an approach to the drought construct by predicting the behavior of the initial projection of the proposed multivariate model. It was necessary to specify a response variable for its development by the intentions and scope of this study. The specific case was to carry out an approximation of the drought construct by predicting the behavior of the amount of available water in the face of the different conditions that could occur during the same time, all through the availability of water in Mexico variable, which could easily represent a causality between the various characteristics of the variables used and the amount of water available in correlation to the various aspects analyzed.

The water availability was considered the dependent variable for the proposed model, and it is contained within the block of variables related to the part of the aquifers. Consequently, the other 25 remaining variables were considered to be independent for the regression model. It should be noted that during this part of the methodological design, the initial approach to the model was carried out to later consider different interactions in order to carry out the optimization of the results and the conditions that verify the assumptions of utility and validation of the model.

*2.8. The Initial Base Model's Progressive Improvement Is Based on the Premises of Linearity, Heteroscedasticity, Collinearity, and Normality of the Errors*

The last phase focuses on optimizing the initial model specified in the previous step through a sensitivity analysis carried out with some statistical treatments on the observations (transformations and elimination of outliers) and the variables' statistical significance. These subsequent model approaches focused primarily on validating the assumptions that accompany the multiple linear regression model to ensure that the projections made through the final version of the proposed model are meaningful.

Among the assumptions contrasted in the approximations materialized by each model generated is the explanatory variables' multicollinearity. That is, if there is some linear dependence between them or if there is a strong correlation between them [48], the main problem of multicollinearity is that the least-squares estimators of the coefficients of the variables in the linear dependencies with this effect come to present variations. Therefore, all the additional adverse effects become a consequence of these significant variations [63]. On the other hand, atypical points also affect the regressor's center, making it biased, indicating heteroscedasticity problems. One way to reduce heteroscedasticity is the logarithmic transformation [45], which minimizes the variability in the data, keeping the variance constant in the errors of the explanatory variables (homoscedasticity) [48].

On the other hand, it has traditionally been assumed that errors are usually distributed in recent years. However, it has also been realized that normal distributions are less frequent in practice [64], although in a generalized way, the consensus remains that random errors are normally distributed, with zero mean and constant $\sigma^2$ variance, which means that they become a fundamental parameter in the regression model [46]. In addition to contrasting the model's assumptions, and once the optimization options were exhausted, a statistical selection process was used to add and eliminate the regressive variables included in the model's final version.

It is essential to underline that the observations in the variables used in this step were transformed and did not contain any atypical observations. Therefore, the application of the step-by-step procedure was significantly facilitated. It, in turn, supposes the entry and exit of variables from a forward and backward perspective according to a level of significance stipulated as 0.05.

## 3. Results

*3.1. The Relationships Were Identified Between the 26 Variables from the Correlation Matrices*

In order to be able to evaluate the latent relationships between the different selected variables in greater depth, a correlation matrix was generated to facilitate the identification of the magnitude of the relationships, and the sense (positive or negative) of the said matrix is represented graphically, based on a color scale, showing the intensity of the inherent correlation between the 26 variables studied.

It is essential to analyze the correlation between the so-called predictor variables and the response variable water availability "A Disp" to understand water availability in Mexico. Figure 3 refers to two versions of the correlation matrix developed. In part (a), a correlation matrix is observed without any ordering or grouping. In part (b), the same conditions are presented except that they are grouped according to the intensity of the relationship and affinity for the tones predicted by the color scale of the graph.

The grouping of colors carried out in Figure 3b gives indications regarding the correlation between the different groups of variables. This is the case, for example, of the extraction volume and natural runoff, total renewable water, and the total average recharge of aquifers that have a significant positive relationship greater than 0.5. Despite the renewable water per capita, the deficit and the total of aquifers barely maintain a relationship, which is less than 0.5.

The significant amount of dark violet tones in the second figure suggests the existence of strong relationships between the predictor variables, which would subsequently generate problems in complying with the assumptions of the model, so it would be necessary to

carry out transformations in the variables in order to eliminate adverse effects caused by strong correlations between variables.

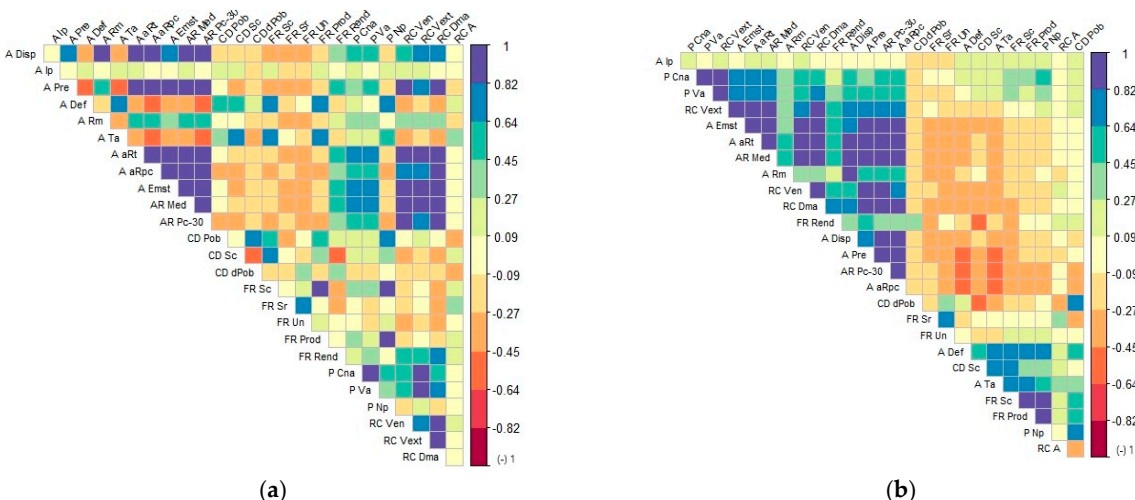

(**a**)                    (**b**)

**Figure 3.** Correlation matrices developed from the interaction of the 26 variables selected and extracted from the SINA records. (**a**) The matrix on the left side represents the correlation of the variables in a non-grouped manner by affording the intensity. The color of the relationship scale indicates it; (**b**) the matrix on the right side is ordered according to the affinity between the intensity of the analyzed variables' relationships.

As already pointed out, there is an inherent link between this group of variables and the different types of droughts and water availability. This condition is exemplified in Figure 4, which intends to demonstrate the intrinsic relationships between the data in Table 1 and drought types.

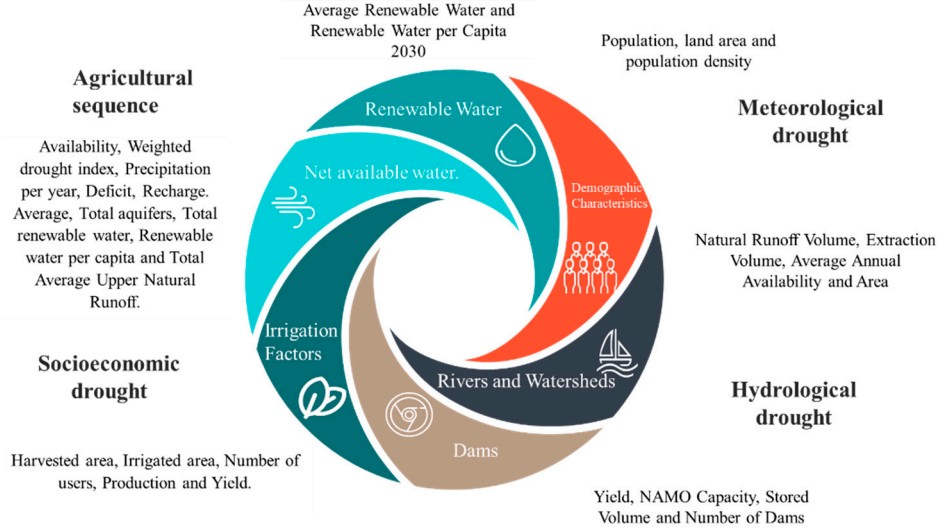

**Figure 4.** Relationship between the set of variables and the types of droughts.

The variables used to reflect the behavior of the RHAs in the country. According to the hydrological regions for the variable, the grouping of the observations reflects the differences between them numerically [52]. These differences are due to geographic conditions, which are mainly due to the climatic diversity in Mexico. Therefore, if the fact that the supply and demand of water per inhabitant is unequal is added to all of this, a situation that is due to various socioeconomic activities, weather conditions, and the availability of water resources, among other factors [10], it is a fact of the enormous dispersion between the observations of each of the variables that make up the model

is evident. The results obtained in a preliminary analysis of the model's development descriptively showed the significant difference between the magnitudes as observed in the histogram generated with the variables related to agricultural drought in Figure 5.

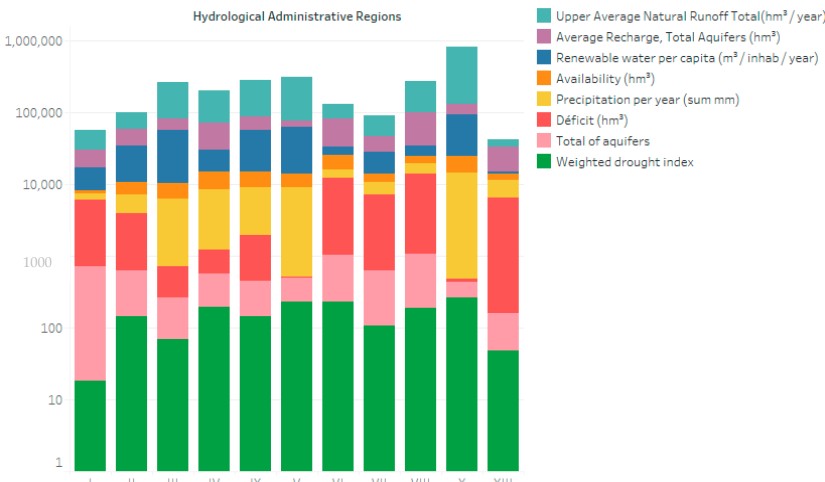

**Figure 5.** Comparative graph of the magnitude of the variables related to agricultural drought using the total sum of the observations transformed into a natural logarithm, with the differences between them presented using the proposed histogram, showing their differences by region and variable.

This simple graphical analysis compares all the model variables' magnitude and dimensions, which is sometimes uneven between regions and variables.

The graph gives indications about the dispersion of the observations obtained and the trend in specific cases. For example, in the Southeastern regions, the variables related to the abundance of rain and water availability tend to present higher magnitudes in their estimates in relation to others. Now, as an introductory part to the development of the model, a scatter plot is presented in Figure 6 with the first five variables (availability, weighted drought index, precipitation per year, deficit, and average recharge). Only these five are presented because the visualization of the 26 variables simultaneously causes the detail of the scattering clouds to be lost, which reduces their understanding. Figure 6 graphically represents the dispersion of the points that the variables generate in relation to water availability; this is seen in greater intensity in the areas where there is a grouping of points about the compared variable.

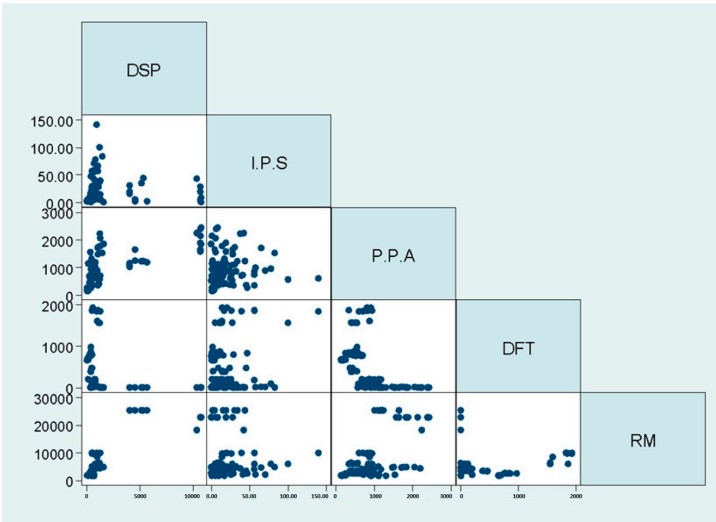

**Figure 6.** Scatter plot for the availability of the variable (DSP), weighted drought index (IPS), precipitation per year (PPA), deficit, and mean recharge (RM).

The case of the quadrant in which water availability shows a scattered cloud with a positive trend linked to the weighted drought index, on the opposite side, is also observed in the deficit compared to the weighted drought index. This comparison of the points indicates a negative trend. Finally, the conditions are also observed where it is not possible to appreciate any relationship. This is the case of the average recharge and the deficit. Another critical point that is considered is the shape of the scattering clouds. In some cases, these form scattered point clouds, for example, between annual precipitation and the weighted index where these points seem to indicate the need to carry out a transformation in the data to avoid problems with the fulfillment of the model's assumptions.

### 3.2. The Development of the Proposed Multivariate Base Model

The series of analyses previously carried out determined the behavior of the variables among the RHAs to establish preconditions for developing the proposed multivariate model. It should be noted that the results of the correlational analysis yielded different versions of the model, which were considered as a series of interactions. However, the first approximation or base model was formed through the 25 predictor variables ($X_i$) selected and the response variable ($Y$) water availability, adding up to the 26 projected variables. To perform the quantitative calculations corresponding to the model, the statistical package STATA version 13 (StataCorp, College Station, TX, USA) was used to facilitate its statistical configuration. In this first approximation, the variable named average renewable water was discarded from the model. It was suppressed because the statistical package determined its contribution to the analyzed model as non-significant. Said variable was no longer used in any of the following approximations. Table 2 shows the output results of the proposed base model, in which each of the output records provided by STATA for the coefficients of the regression equation and their statistical significance can be observed in detail. In this initial version of the model, no transformation or adjustment was applied to the variables used.

**Table 2.** Coefficients ($\beta$) and their statistical significance values ($p$-value, 5% and 1%) corresponding to the proposed base model about 25 related predictor variables between RHAs.

| (1) Availability | (2) Coefficients ($\beta$) from the Regression Equation | (3) $p$-Value | (4) Significance of the Coefficient at 5% | (5) Significance of the Coefficient at 1% |
|---|---|---|---|---|
| Weighted drought index | −0.8053803 | 0.638 | | |
| Precipitation per year (mm) | −0.0752947 | 0.756 | | |
| Deficit (hm$^3$) | −0.0021905 | 0.993recharge | | |
| Average(hm$^3$) | −1.226786 | 0.689 | | |
| Total, from aquifers | −4.085112 | 0.408 | | |
| Total renewable water (hm$^3$/year) | 1.328022 | 0.664 | | |
| Renewable water per capita (m$^3$/inhabitants/year) | 0.4585418 | 0.024 | 0.459 (2.30) * | |
| Average surface natural runoff total (hm$^3$/year) | −1.350149 | 0.659 | | |
| Renewable water per capita by 2030 (hm$^3$/year) | −0.0885943 | 0.681 | | |
| Population (inhabitants) | 0.0000391 | 0.402 | | |
| Continental surface (km$^2$) | 0.0039529 | 0.04 | 0.04 (2.09) * | |
| Population density (inhab./km$^2$) | 0.2932887 | 0.702 | | |
| Harvested area (ha) | 0.0001496 | 0.869 | | |
| Total irrigated area (ha) | −0.0006992 | 0.469 | | |
| Number of Users | −0.0059074 | 0.072 | | |
| Agricultural production (thousands of tons) | −0.0081616 | 0.816 | | |
| Yield (tons ha) | 19.45961 | 0.063 | | |
| NAMO Capacity (hm$^3$) | 0.0033639 | 0.91 | | |
| Stored Volume (hm$^3$) | −0.0009924 | 0.965 | | |
| Number of dams | −15.57708Esc | 0.13 | | |
| Vol. annual nat. (hm$^3$) | −0.0620822 | 0 | −0.062 (5.59) * | −0.062 (5.59) ** |
| Extraction(hm$^3$) | volume0.0403797 | 0.003 | 0.040 (3.04) * | 0.040 (3.04) ** |
| Average annual availability of rivers (hm$^3$) | 0.0161765 | 0 | 0.016 (3.68) * | 0.016 (3.68) ** |
| Area (km2) | 0.0013894 | 0.361 | | |
| _cons | −1251.237 | 0.015 | | |

\* $p < 0.05$; \*\* $p < 0.01$.

In general, the base model, which has also been named model A for differentiation purposes to the other versions projected from it, presents a series of critical statistical coefficients; these have been summarized in Table 3 These factors show the independent variables' explanatory power concerning the response variable, the mean square error obtained for the line's fit, and the model's statistical significance with the beta coefficients generated.

**Table 3.** Representative statistical measures of the proposed regression model (A).

| Measure Analyzed | Result Obtained |
|---|---|
| Number of obs | 104 |
| Prob > F | 0.0000 |
| R-squared | 0.9927 |
| Adj R-squared | 0.9905 |
| Root MSE | 274.33 |

According to the values provided in Table 3, model A has a coefficient of determination of 0.9927, which assumes that 99.27% of the time, the variability of water availability is explained linearly by the predictor variables. On the other hand, the adjusted coefficient of determination is 0.9905. This value will be beneficial later as it will serve as a reference to compare the fit of the other versions of the model. In general, the model's statistical significance yields a value of 0.0000, which shows that at least one of the beta coefficients obtained in Table 2 causes some effect on the response variable, assuming that the regression is generally significant. The regression equation's coefficients' effect is evaluated in a particular way in columns 4 and 5 of Table 2. In these columns, the significant beta coefficients have been placed at a level of 5% and 1%, respectively. Finally, the error's mean square determines the fit of the multiple equations concerning the real values. In this case, this value is equal to 274.33, which is a considerably high value.

*3.3. The Progressive Advance of the Base Model to Other Versions with Better Explanatory Conditions*

Once the premises of model A were analyzed with the multiple regression's statistical assumptions, it was necessary to make adjustments concerning the observations' dispersion. It essentially caused a few significant coefficients in the regression equation's multiple and the multicollinearity and heteroscedasticity, evidenced between the variables and the model's residuals while considering their lack of normality. The few significant coefficients in the regression equation are very evident once the data in columns 4 and 5 in Table 2 are carefully analyzed. In these, it is perceived that only five variables are significant for the regression equation at a significance level of 5% and only three at a level of 1%; this means that only 20% and 12%, respectively, of the variables that were analyzed explain the availability of water in the base model. The few significant variables, with the transgression of the model assumptions, implied the use of techniques such as the transformation and reduction of variables to improve the conditions of the proposed model and to be able to reduce the effects of dispersion between the observations of the selected variables.

This adjustment of the dispersion caused by the atypical data of the observations belonging to the variables led to the development of three more versions of the base model, denoted with the letters B, C, and D. On the other hand, the last two versions use variables transformed to a natural logarithm to correct the transgressions of the assumptions caused by the data by standardizing them.

Following the data obtained from the adjustments made to reduce the model variables' dispersion, Table 4 could be formed; it compares the assumptions analyzed in each of the different interactions represented by the models developed. This comparison provides details about the number of observations of the analyzed model, the normality of the residuals based on Shapiro Wilk's tests, Shapiro France, Skewness, and the multicollinearity test, and the diagnosis of heteroscedasticity using the Breusch–Pagan and Ramsey tests.

**Table 4.** Summary of the validated assumptions for models A, B, C, and D.

| | Assumptions of the Regression | | |
| --- | --- | --- | --- |
| Model Proposed | Normality of the Residuals | Multicollinearity | Heteroscedasticity |
| A (104 obs.) | ✗ | ✓ | ✓ |
| B (88 obs.) | ✓ | ✓ | ✓ |
| C (80 obs.) | ✓ | ✓ | ✗ |
| D (75 obs.) | ✓ | ✓ | ✗ |

✓ Presents conditions, ✗ Do not present conditions.

The comparatives summarized in Table 4 are derived from the transformations carried out in the model variables in order to be able to structure a version much more robust, which in turn is reflected in the estimates of water availability generated from the regression equation of each model evaluated.

On the other hand, the conditions of model D already presented a significant advance with the multiple regression's statistical assumptions. However, there were still perfectible conditions, for example, the multicollinearity between the variables and the small number of significant beta coefficients in the regression equation established by it. This is noticeable once we consider the number of significant variables in the model since they barely ranged between 10 and 5 variables at a statistical significance of 5% and 1%, respectively, so only between 40% and 20% of the variables in the model were considered for water availability.

The importance of one or another variable within the structure of the proposed model became a fundamental aspect of improving its conditions; this is when we consider that the introduction or exclusion of a variable in a regression model only can be justified by reason and the literature (which do not always coincide) and never because there is a correlation in practice since this can be spurious [65].

Based on this fact, it was necessary to use a discriminant method to consolidate the variables that would configure the model once the other methods, such as eliminating outliers and the variables' transformations, had been exhausted. Thus, a step-by-step selection of variables was necessary to increase the number of significant coefficients in the prediction equation for water availability. Simultaneously, the model was improved, making it even more explanatory and robust to the response variable. The sequential search methods have a commonness in the estimations of regression equations with a set of variables to later selectively add or eliminate variables until achieving some joint criterion measure; these methods use the construction of a statistic called partial F, which contrasts the partial Fs in two models that are intended to be used to make a selection of variables [49,50].

For making this selection of relevant variables for the model, the complementary functions of STATA were used to introduce and discard variables through the comparative analysis of the Fs in order to generate the necessary iterations and select those with a *p*-value (0.05) maximum (forward) or to discard those that do not have a minimum *p*-value (0.05) (backward).

The results were two more approximations of the model called E and F. For both cases, the number of significant variables about the response variable water availability increased considerably compared to the previous versions. Table 5 shows the results obtained concerning the number of variables considered in the model and the significant increase in both cases.

Table 5 summarizes the models' most relevant information, and it highlights the number of significant variables for the regression concerning a significance level of 5% and 1%, respectively.

Table 6 contains the elements generated for model F, representing the final version of the proposed model. Said version comprises 16 variables, which were selected through a procedure where all the variables are introduced initially, and later those that lose their significance are extracted. Likewise, Table 7 establishes a comparison between the

statistical assumptions of models E and F. The data presented are identical to that made with models A, B, C, and D to compare the models' assumptions. As can be seen, the results are noticeably better for model F, so this was considered the final interaction of the proposed model's analysis for the process. Together, these results provide a broad overview of the progressive interactions made from the different models (A, B, C, D, E, and F) in order to be able to determine a version as close to the assumptions of the model multiple linear regression.

**Table 5.** Summary of models E and F to the number of significant variables according to 5% and 1% levels, respectively.

| Model | Number of Variables Included | Variables Significant at 5% | Variables Significant at 1% |
|---|---|---|---|
| E | 14 | 12 | 10 |
| F | 16 | 16 | 12 |

**Table 6.** Coefficients and their statistical significance values (*p*-value, significant at 5% and 1%) referring to model F from the backward perspective.

| Availability | Coefficient (*β*) from the Regression Equation | *p*-Value | Significance of the Coefficient at 5% | Significance of the Coefficient at 1% |
|---|---|---|---|---|
| Weighted drought index | 0.0264151 | 0.003 | * | ** |
| The average annual availability of rivers (hm$^3$) | 0.0705061 | 0.003 | * | ** |
| Deficit (hm$^3$) | −0.1481127 | 0 | * | * * |
| Average(hm$^3$) | Recharge 0.955508 | 0 | * | ** |
| Extraction(hm$^3$) | volume−0.2919614 | 0 | * | ** |
| Total renewable water (hm$^3$/year) | −8.776252 | 0.007 | * | ** |
| Renewable water per capita (m$^3$/inhab/year) | 8.669164 | 0.007 | * | ** |
| Area (km$^2$) | −0.0567943 | 0.038 | * | |
| Stored Volume (hm$^3$) | −0.0956543 | 0.028 | * | |
| Number of dams | −0.2807762 | 0 | * | ** |
| Continental surface (km$^2$) | 7.691091 | 0.015 | * | |
| Population density (inhab./km$^2$) | 8.247614 | 0.01 | * | ** |
| NAMO Capacity (hm$^3$) | 0.4836722 | 0 | * | ** |
| Total irrigated area (ha) | −0.2023765 | 0.002 | * | ** |
| Number of Users | 0.1036336 | 0.048 | * | |
| Agricultural production (thousands of tons) | 0.0979167 | 0.009 | * | ** |
| _cons | −105.9004 | 0.015 | * | |

* $p < 0.05$; ** $p < 0.01$.

**Table 7.** Comparison of statistical assumptions for models E and F generated from a step-by-step perspective.

| | Assumptions of the Regression | | |
|---|---|---|---|
| Model Proposed | Normality of the Residuals | Multicollinearity | Heteroskedasticity |
| E (stepwise forward 75 obs.) | ✓ | ✓ | ✓ |
| F (stepwise backward 75 obs.) | ✓ | ✓ | ✗ |

✓ Presents conditions, ✗ Do not present the conditions

The conditions analyzed in Table 7 refer to the fulfillment of the linear regression model's statistical assumptions; as shown, version E fulfills all the assumptions. However, the significant variables determined by the significance tests compared to the coefficients of beta generated from them are lower than those obtained in the F version or final model. For

its part, the F model presents heteroscedasticity conditions, but the number of significant variables is greater than the previous one, for which it has been decided to opt for this version despite the non-compliance to the constant variance in the residuals.

On the other hand, Table 8 is a general summary of the regression results in model E, which contains fewer variables than version F. However, it should be remembered that it is completely attached to the parameters of the assumptions of normality, collinearity, and homoscedasticity.

**Table 8.** Coefficients and their statistical significance values (*p*-values, significant at 5% and 1%) referring to the model developed from the forward perspective, introducing significant variables to the model from a *p*-value less than 0.05, model E.

| Availability | Coefficients ($\beta$) from the Regression Equation Recharge | *p*-Value | Significance of the Coefficient at 5% | Significance of the Coefficient at 1% |
|---|---|---|---|---|
| Average($hm^3$) | 0.8696483 | 0 | * | ** |
| Deficit ($hm^3$) | −0.1020095 | 0 | * | ** |
| Stored volume ($hm^3$) | −0.0528958 | 0.219 | | |
| Vol. extraction (hm3) | −0.2832898 | 0 | * | ** |
| area Harvested(ha) | 0.0201653 | 0.559 | | |
| Number of dams | −0.3176423 | 0 | * | ** |
| Continental area ($km^2$) | −0.549637 | 0 | * | ** |
| NAMO capacity ($hm^3$) | 0.4871519 | 0 | * | ** |
| Renewable water per capita ($m^3$/inhab./year) | 0.3211761 | 0 | * | ** |
| Area (km2) | −0.1068207 | 0 | * | ** |
| The average annual availability of rivers ($hm^3$) | 0.1502486 | 0 | * | ** |
| Weighted drought index | 0.0201989 | 0.024 | * | |
| Total average surface natural runoff ($hm^3$/year) | −0.4389393 | 0.002 | * | ** |
| Total irrigated area (ha) | −0.0774245 | 0.022 | * | |
| _cons | 7.8723 | 0 | * | ** |

* $p < 0.05$; ** $p < 0.01$.

## 4. Discussion

This study demonstrates the progressive development of a multivariate model to predict water availability by interacting with the different variables configured with the data from the records concentrated in the national water information system platform. It should be noted that the model is configured based on a progressive approach. Each version of the model must be taken into account as an interaction. It has the purpose of improving the statistical conditions of the results obtained. Identifying the different versions seeks to explain the different perspectives that could arise between this set of variables intrinsic to drought and its repercussions on water availability and the forecast of the conditions during the different types.

On the other hand, it is important to underline some considerations with the different versions obtained. In the case of the first two models, standard observations without transforming were used. However, for all other versions, it was necessary to carry out the data transformation by the method natural logarithm; this facilitated the standardization of the regression residuals, further improving the detection of outliers in the variables.

In Table 9, a summary of the principal regression coefficients obtained for each model is presented in order to more widely understand the interactions that were developed.

These results have important implications that significantly influence the capabilities of each model, the first of which is the significance of the proposed model (3). The *p*-value of 0.0000 remains constant for the different versions, which shows the importance of the variables used in the model; this makes it reasonable to think that at least some of the variables of the analyzed set infer and explain the availability of water under the different

conditions provided by them. Now, the magnitude of the variability explained by the predictor variables is defined by the determination coefficient (4) This always remained high in all the generated approximations, whereby the said value was projected above 99%. This was conclusive to the extent that the different adjustments to the dataset were developed, and even despite the application of the discriminant techniques of inclusion and exclusion of variables proposed in the previous section, this made it possible to show that the value remained constant under the different modeling conditions for the various variables analyzed. Said coefficient obtained its highest value in model B with 99.81%, which means that the predictor variables explained 99.81% of the variability present in the availability of water. Regarding the adjusted determination coefficients (5) that were considered indicators of the best fit between the different versions of the model, 99.95% of model C stands out, where the number of observations of the model seems to cause a significant effect on its coefficient value that focuses on the best fit.

**Table 9.** General summary of the regression correlations obtained in each of the models formulated and analyzed.

| Summary of the Correlation Coefficients of the Models | | | | | |
|---|---|---|---|---|---|
| Model (1) | Number of Obs (2) | Prob > F (3) | R-Squared (4) | Adj R-Squared (5) | Root MSE (6) |
| A | 104 | 0.0000 | 0.9927 | 0.9905 | 274.33 |
| B | 88 | 0.0000 | 0.9981 | 0.9973 | 123.75 |
| C | 80 | 0.0000 | 0.9927 | 0.9995 | 0.10172 |
| D | 75 | 0.0000 | 0.9963 | 0.9945 | 0.07615 |
| E (Forward) | 75 | 0.0000 | 0.9950 | 0.9938 | 0.08066 |
| F (Backward) | 75 | 0.0000 | 0.9956 | 0.9944 | 0.07711 |

For models C, D, E, and F, transformed variables were used to natural logarithm, so the error is significantly lower in the first two and is not comparable.

On the other hand, and considering the results found, the evident constant multicollinearity between the same variables stands out. In the end, the mean square of the error (6) is comparable between models A and B since their magnitudes are not transformed for normalization, as in the case of the remaining models (C, D, E, and F). Therefore, the best fit between the first two models turns out to be B due to its smaller magnitude. Finally, model D has a minor magnitude, so it assumes the best fit of the line projected from it.

This combination of results provides different elements to analyze the perspectives derived from the interactions carried out. On the one hand, model F has been considered the final version mainly due to the number of significant beta coefficients, causing it to be the model with the largest number of relevant variables to explain water availability.

In summary, the F model adequately assumes conditions such as the normality between the residuals; however, it presents multicollinearity between the predictor variables, although it should be emphasized that there is no heteroscedasticity between its residuals and that in general, the variables that comprise it are considered to be significant at 5%, so they all infer the availability of water. The beta coefficients for each of the equation variables are vital to highlight the sign's magnitude so that a coefficient with a negative sign causes an unfavorable effect on water availability. These are the deficit variables, namely extraction volume, total renewable water, area, stored volume, number of dams, and irrigated area.

An unexpected finding is that the variable renewable water and area present a negative coefficient when analyzing the condition of the total renewable water, which represents the amount of fresh water available in lakes, rivers, and underground aquifers from the rains and is currently managed through dams and absorption wells that infiltrate the rainwater aquifers artificially. In that case, it is easy to intuit that the volumes destined for their conservation are deducted from the available records for the different uses, which confers its negative nature in the area where more excellent geographic coverage reduces water availability.

It should be noted that the weighted drought index, average annual availability in rivers, average recharge, renewable water per capita, continental surface, population density, NAMO capacity, number of users, and agricultural production are positive; therefore, it is considered that an increase impacts in such a way that is favorable to the availability of water.

On the other hand, in the case of population density and the number of users, the data interpretation is complicated since both variables should have a negative coefficient; however, the conditions are different in some models. This effect of finding positive coefficients in variables that adversely impact availability supposes very particular conditions.

This effect seems to change from model to model. Instead, in model B's coefficients, these two variables have a negative sign, which is understandable because both variables refer to water consumption and not its recharge. However, in the other models' interactions, it can be seen that the effect of the coefficient is positive. The change in conditions from one model to another could be related to some effects such as constant multicollinearity.

It can be shown that certain variables would be redundant for the model, for example, in the number of users and the population density. On the other hand, the drought index's coefficient also presents a positive value, which is not very logical if we consider the fact that natural resource availability is considered in a drought. However, the approach is different since the increase in drought also requires an increase in the liquid availability depending on the coverage of needs from the different sectors where this exact cause affects.

To further understand how the F model relates the variables, Figure 7 presents a graphical analysis where each variable is represented so that it is possible to detail the slope of the variable for water availability. One can also identify the scattering cloud generated around the data fit line. This favors a better understanding of the beta coefficients in the line's equation and the contribution and direction of the slope generated by it.

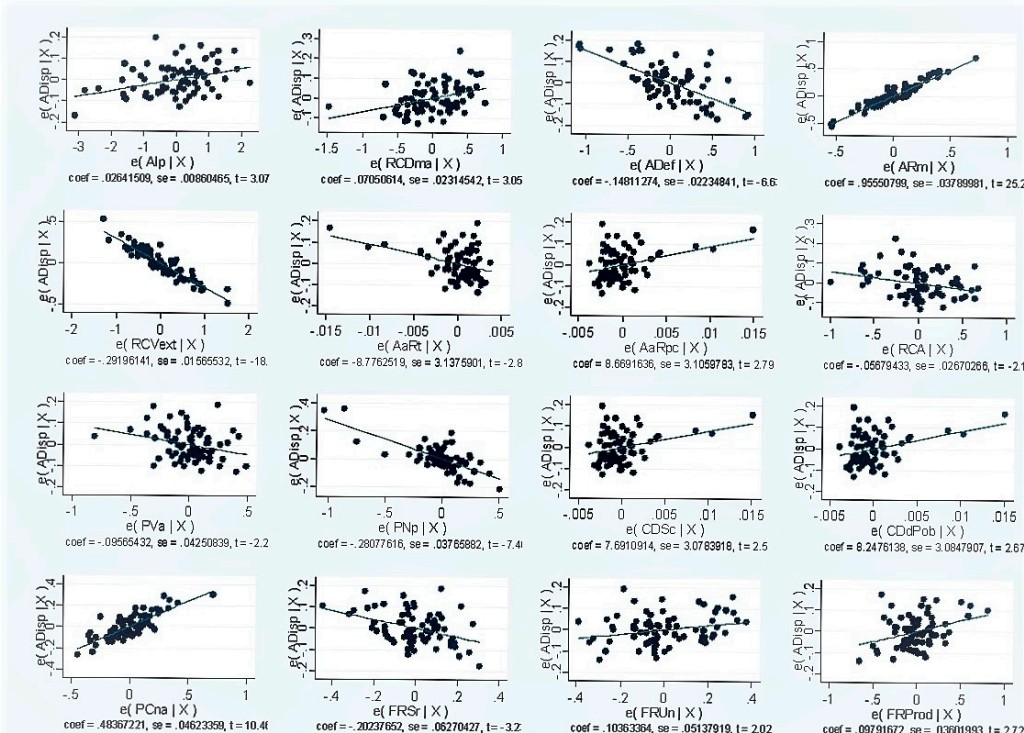

**Figure 7.** Partial regression graph for each of the independent variables.

However, the improvement of the assumptions in the model is evident once the changes that occurred in the variance and multicollinearity of the residuals in each interaction are compared since the output conditions in the initial base model are comparatively very distant from ideals. Moreover, the progressive advance of the versions is evident if the

assumptions of each one of them are contrasted. Thus, in a particular way, the progressive improvement of said elements can be appreciated in detail in summarizing the contrasting results as shown in Table 10.

**Table 10.** Comparative analysis of the regression model's assumptions for the residuals of the different interactions carried out.

| Proposed Model | Multicollinearity | Heteroskedasticity Homoscedasticity | Residual Graph |
|---|---|---|---|
| A (104 obs.) | All VIFs are > 10. Only the weighted index variable is the only one whose value is within an acceptable range. There is multicollinearity in the | Breusch-Pagan/Cook-Weisberg test for heteroskedasticity Ho model: Constant variance = 0.000 There is heteroskedasticity in the model |  |
| B (88 obs.) | All the VIFs are > 10. Only the variable of the weighted index is the only one whose value is in an acceptable range. There is multicollinearity in the | Breusch-Pagan/Cook-Weisberg model test for heteroskedasticity Ho: Constant variance = 0.0254 There is heteroskedasticity in the model |  |
| C (80 obs.) | All the VIFs are > 10. Only the variable of the weighted index is the only one whose value is in an acceptable range. There is multicollinearity in the | Breusch-Pagan/Cook-Weisberg model test for heteroskedasticity Ho: Constant variance = 0.1032 There is no heteroscedasticity in the model |  |
| D (75 obs.) | All VIFs are > 10. Only the variable weighted index is the only one whose value is in an acceptable range exists multicollinearity in the model | Ramsey RESET test using powers of the fitted values Ho: model has no variables omitted = 0.1636 There is no heteroscedasticity in the model |  |

**Table 10.** *Cont.*

| Proposed Model | Multicollinearity | Heteroskedasticity Homoscedasticity | Residual Graph |
|---|---|---|---|
| Forward | Some values of the VIF are > 10. Except for the variables number of dams, area, average recharge, and the weighted drought index. There is multicollinearity in the | Breusch-Pagan/Cook-Weisberg test for heteroskedasticity Ho model: Constant variance = 0.0089 The model shows heteroscedasticity. Ramsey RESET test using powers of the fitted values Ho: model has no omitted variables = 0.0207 |  |
| Backward | Some VIF values are > 10. The variables weighted indexes, extraction volume, mean recharge, and area are in an acceptable range There is multicollinearity in the model | RamseyRESET test using powers of the fitted values Ho: model has no omitted variables = 0.0002 The model does not show heteroskedasticity of agreement with the first test Ramsey RESET test using powers of the fitted values Ho: model has no omitted variables = 0.0985 |  |

Although the indicator of the Variance inflation is a notoriously high factor in the first models, it tends to decrease in the final versions, as does heteroscedasticity, as can be seen in the results and the residuals graphs in Table 9.

It should be noted that the effect of normality between the residuals and the different statistical tests developed that involve this effect had the implication of various techniques and the discriminant analysis of the variables and the normality of the residuals. These elements are summarized in Table 11, which concentrates on the results of the models. The transformation of the results in each version is noticeable from the reduction of observations, transformations, and the discriminant analysis performed between the variables, which changes the shape of the density distribution markedly for each of the cases analyzed.

In turn, this study's limitations are due to a greater extent to the bias of the data that directly affect the significant differences between the RHAs. Furthermore, the constant multicollinearity is due to the correlated and redundant variables in the model versions, reflected as adverse effects.

It should be noted that there is abundant space to continue advances in the determination of the appropriate way to group variables concerning the nature of their numerical characteristics. The use of other multivariate techniques such as the analysis of primary factors and clusters could add new evidence about how the variables can be assigned so that there are other versions of the model that are more focused on particularized questions regarding the variable water availability, as well as further characterizing the effects of the types of droughts based on the aspects evaluated. It is also essential to consider the distinctions between the different types of droughts since the elements of the physical processes that cause each one is different and sometimes do not entirely coincide, creating variations that are potentially contradictory to the moment's intuition [35].

**Table 11.** Comparative summary of the different models proposed based on the normality conditions in the residuals.

| Proposed Models | Graph of Normality of the Errors P. | Graph of Normality of the Errors Q. | Graph of the Density of the Errors. |
|---|---|---|---|
| A (104 Obs.) |  |  |  |
| B (88 Obs.) |  |  |  |
| C (80 Obs.) |  |  |  |
| D (75 Obs.) |  |  |  |
| Forward |  |  |  |
| Toward Backward |  |  |  |

Finally, among the future and possible research questions formulated from the results obtained in this research are how a model of this nature can be integrated into humanitarian strategies in the face of different drought conditions. The model's usefulness regarding humanitarian aid issues and the relief of suffering due to water availability have been primarily considered in the analyzed model.

This article focuses on supposing a change of perspective about the modeling of drought and its climatological implications commonly addressed in other publications. The model's orientation is water availability and how it is affected by different factors linked to drought's distinct components. The conformation of a model prediction of water availability is particularly noticeable due to the different perspectives addressed for its conformation. Through interactive and meticulous analysis, details arise concerning the relevance of the variability in the observations and their regionalization, in addition to the inclusion and exclusion of some variables in it. One is able to configure a different understanding of drought from a response variable such as water availability.

## 5. Conclusions

This research focused on the development of a model to determine water availability from a set of variables related to the characteristics present in the different types of droughts, in order to support water management through strategies focused on counteracting the effects of drought among the different sectors that are affected by this phenomenon.

The results identified propose different conditions of importance to improve the understanding of water availability at the national level, so it is clear that not all the selected variables were necessary. However, initially, their selection seemed to be the most appropriate due to their close link with the national water system. For example, this is the case of the average renewable water variable, which was discarded from the first approximation of the model. Now, the diverse nature of the different hydrological regions strongly influenced generating an excellent dispersion between the observations, which brought with it inconveniences in the fulfillment of the assumptions of the projected models. Therefore, the transformation was carried out through use of the data, the selection of variables by significance, and the reduction of outliers to eliminate the repercussions of these effects.

On the other hand, the contrasts carried out employing the versions of the proposed gave it different meanings to the variables used and their link with the water availability variable. These findings are of great help, considering that the more the cause-and-effect relationships caused by the drought can be explained, the more progress can be made to understand the details of a phenomenon with a great complexity such as this one.

It should be noted that some variables that seem to be related are of utmost importance to establish analysis groups, which improve the understanding of drought actions. Identifying the positive repercussions, negative or neutral as the case may be, would also have a positive impact on the development of plans and programs, in order to be able to establish a much more precise guide to the possible impacts, as the conditions of the analyzed variables change.

The grouping of blocks of variables that seem closely related has been one of the elements established with great notoriety in the results presented in the matrix analysis between variables. This correlation assumes a pattern of relationships that various multivariate techniques in subsequent studies could later confirm.

Regarding the relationship between the variables, the research has shown that the set of variables is made up of the capacity in NAMO, stored volume, extraction volume, renewable water, average runoff, total area, harvested area, agricultural production, number of dams, precipitation per year, and average annual availability of rivers. These reflect a positive relationship between themselves, which suggests the possibility of considering them a specific group due to the affinity between them. On the other hand, the agricultural yield, precipitation, renewable water per capita, population density, continental surface, deficit, and the number of aquifers present opposite conditions to the previous ones. The

interaction between this set assumes negative-type relationships. In general, the variables' inherent nature manages to infer the type of relationships between them, which essentially specifies how their values increase and decrease. For example, as the rainfall variable increases, water availability in rivers and basins increases. In contrast, the less renewable water per capita, the larger the deficit variable. In general, the results and conclusions derived from this research can help establish plans and actions that improve the availability of water in Mexico under extremely complex conditions emanating from the phenomenon of drought or due to partial effects on the variables that make up the model.

Developing regions or indices based on other multivariate statistical techniques could give continuity to the model developed in this article, creating particular analysis conditions for groups of specific regions. Alternatively, delimiting the information by sectors or administrative hydrological regions would generate a less general and more specific understanding of water availability, linked to the resources and conditions represented by these new interactions, which would be the result of the application of other multivariate techniques, under conditions not as generalized as those currently proposed by the model. Finally, this research has examined the data provided by the variables regionally, which cause significant differences between some of them. This makes it necessary to consider the study's crucial limitations, such as the redundancy of some variables, evidenced by the constant multicollinearity in the different versions of the model, the atypical observations in the variables, and the availability of the variables records in the SINA platform.

**Author Contributions:** D.S.-P. and H.R.-M. conceived the idea of the research together with J.-L.M.-F., P.C.-O. mainly corrected the style and structure of the article, and H.R.-M. was particularly involved in each of the parts of the research. D.S.-P. conceptualized the theoretical framework, the discussion and development of the article, while J.-L.M.-F. developed the numerical analysis and supervised the data processing. Finally, D.S.-P. has also been involved in obtaining funds through the Universidad Popular Autónoma del Estado de Puebla. All authors have read and agreed to the published version of the manuscript.

**Funding:** This research was funded by the National Council of Science and Technology CONACYT, and the Universidad Popular Autónoma del Estado de Puebla.

**Institutional Review Board Statement:** Not applicable.

**Informed Consent Statement:** Not applicable.

**Data Availability Statement:** The data supporting the reported results can be requested with the corresponding author hugo.romero@upaep.edu.mx.

**Conflicts of Interest:** The authors declare no conflict of interest.

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
