# Peer review of "Development of a Multivariate Model Focused on the Analysis of Water Availability in Mexico"

_water, doi:10.3390/w13131779_

Round 1

Reviewer 1 Report

Main overview and comment:

The topic is certainly interesting, and within the scope of the journal. I have, however, some substantial concerns about the paper as it is presented - to name the most important issues. The manuscript presents a critical analysis and proposal aiming at multivariate modelling of droughts using copulas and meta-heuristic method. The academic significance of the current study could be clearly stated in Introduction section. The manuscript does not answer the question raised in its Introduction in relation to managing drought risk in a changing climate. I did not find clear answer in conclusion for thesis stated in Introduction part for example the role of national drought policy.  I recommend to the Authors to change the general idea of the manuscript, or clearly state the answer in the Conclusion section. This must be clarified. The article, however, is not ready for publication but suffers from major drawbacks.

I recommend re-submission for review with correction of grammatical errors by native English speaker, or agent and economic consideration of this specific study. Only some are mentioned in the following.

Specific remarks:

            The article focuses on experimental approach aiming at hydro-meteorological parameters which cause drought preparedness. However, the manuscript suffers from various deficiencies, e.g. some parts of the explanations were not convincing, and some important data is missing to evaluate the mechanisms based on model E and F. Before the acceptance of publication, there should be some solid theoretical evidence provided to support your proposed method was better for water management which should lead to some scientifically significant results. These changes must be made before re-submission:

  1. Abstract must be rewritten in a more precise style. It should be expanded to include the important results (how good/ significant is the data comparing with other similar studies etc.).
  2. The academic significance of the current study could be clearly stated in Introduction. The analytical methods and their hydro-meteorological analysis should be provided in details. More detailed discussion of factors affecting the new drought policy with special reference must be included. Make every attempt to improve the discussion by critically analyzing your findings. It would be excellent, if the important of this issue ware validated by detailed research and thoroughly documented data.
  3. Please carefully check recent literature, and discuss/cite as you see fit and update your reference list according to drought in a human-modified area.
  4. Certain relevant information of the experimental work or results is not given. Samples size used in the study should be provided in table form. Please mention, how many samples were taken and replication (hydro-meteorological parameters) in Tables and Figures. Error bars, or SD might be also provided (if appropriate) reframing drought definition.
  5. Conclusions should be amended to incorporate a broader discussion of the potential application for a nonparametric multivariate multi-index drought monitoring framework, multivariate drought frequency analysis or identification of homogeneous hydrological regions through multivariate analysis.

Constructive feedback:

Material and Methods section are sufficiently explained. The analytical methods and their quality assessment should be provided in details. However, as it stand, the paper aims at a causality which is not feasible to establish, and comes, hence, to conclusions that cannot be drawn from the data-set as currently presented. This is confusing and somewhat misleading. Consequently, the background and rationale of this study becomes not clear referencing to drought structure based on a multivariate standardized drought index. Thus, the novelty of this work is (again as presented in the abstract) questionable, or even absent. This must be substantially reworked from my perspective. I have struggled to fully understand the paper due to the manuscript's poor English. Nonetheless, I think that the document can provide some scientifically novel information, if the major concerns above will be satisfactorily addressed. However, for now, I cannot recommend this work for publication in this journal at is.

            The novelty in research can be defined in many ways. This particular research study overall contribute to the knowledge base of drought mitigation and multivariate assessment and attribution of droughts. Authors using an assumed theoretical framework and this affect the conclusions they draw. As per your description, you have done wide analysis of a well-known method. Please expound the innovation of the study. Is Mexico vulnerable to Mega-drought? It does not necessarily entail inventing a new method or technique. Explain how this analysis will be helpful to image processing, and what it will add to the existing literature.

The topic is not loosely connected with multivariate design of socioeconomic drought and impact of water reservoirs. This gives the impression of a "piecemeal" of randomly selected information and cited references, dealing with case-specific findings, but not with the overall state-of-knowledge. Consequently, the background and rationale of this study becomes not clear, especially modelling month hydrological drought probabilities. The aims are poorly derived and unclear in relation to drought and water shortages in Mexico.

The Conclusion is first of all a compilation of statements, mixed with knowledge from the literature.  The information delivered from Figures 1 and 2 is mostly not sufficiently explained in the text as authors describe a multivariate approach for persistence-based drought prediction. The language is acceptable, but many small errors in grammar, orthography, use of terms and style must be corrected. It is strongly suggested to ask an experienced colleague, or a professional consultant (for example expert in water management) for help in language editing.

Summary of the paper:

            It is my view that the study reported in this manuscript has potential to contribution to knowledge in this area of study, and robust methods were used to conduct research. The authors present a study and investigated drought characterization from a multivariate perspective. However, vulnerability and policy relevance to drought in the semi-arid region for conducting drought-specific risk assessment to increase drought mitigation quality has been investigated since decades. Yet, still new applied research is welcome to solve the still not satisfactorily taken problem. Overall, it seems as if the paper was rushed, and that the authors should to spend more time on improving the language and flow of the manuscript. Therefore, the manuscript needs many improvements before publication.

Author Response

The indicated corrections have been made in the body of the article, so that it has been reformatted once again but considering each of the conditions pointed out about the conditions of improvement for the attached publication. 

Reviewer 2 Report

The manuscript concerns the important issue of the multivariate model focused on forecasting water availability in Mexico. The variables related to aspects such as net available water, renewable water, demographic characteristics, rivers and basins, dams, and irrigation factors were considered in the period of analysis between 2010 and 2017. The following suggestions should be referred to.

It is recommended to highlight the new contributions of the presented research. The general framework of the presented study should be presented. The manuscript is prepared in poor form. Please check the upper script of the units, eg. line 307: units of natural runoff volume; line 309: units of volume of abstraction; line 310: units of average annual availability of rivers and basins. The double numbering of references. Line 152. What is the difference between the references. Why Authors did not analyse only these regions, which do not have enough water, such as the northern and central states: Durango, Baja California, Chihuahua, Coahuila, Nuevo León, San Luis Potosí, Aguascalientes, and Zacatecas?

Something is lacking in the sentence: According to some authors [24]; Line 634: Add some details. What lessons should authorities draw from this analysis? Are there concrete steps that can be recommended and how generalizable are the findings? Can they be applied to other areas? How dependent are they on the specific characteristics of the region under examination? This should be discussed in the point concerning the discussion of the results. Therefore, I propose to consider the possibility of completing the last point for a discussion on the possibilities and limitations of the use of such analysis.

Author Response

(The authors gave the same response as above.)

Round 2

Reviewer 1 Report

Pretty well done and prepared answers. Thank you for your reply.

Author Response

Thank you

Reviewer 2 Report

Line 564: Low quality of the following figure: Figure 5. Comparative graph between all the variables using the sum of the observations to later compare them in the proposed histogram where their differences by region and by variable are revealed.

More details about references should be included in the text (lines: 44-54).

Figure 6. Lack of descriptions of abbreviations.

Line 831. Font of the fig. 11 should be bigger: Table 11. Comparative summary of the different models proposed based on the normality conditions in the residuals.

Add some information about future work.

Round 3

Reviewer 2 Report

Accept in the present form.